# Testing for Hepatitis B and C virus and HIV in mental healthcare settings in England between 2015–2021

Matthew Hibbert[1,2]*, Ruth Simmons[1,2], Peter Dearman[1], James Lester[1], Annabel Powell[1], Cuong Chau[1], Clare Humphreys[1], Liz Hughes[3], Margaret Heslin[4], Monica Desai[1,2], Caroline Sabin[2,5]

**1** Blood Safety, Hepatitis, Sexual Health and HIV Division, The UK Health Security Agency, London, England, United Kingdom, **2** National Institute for Health and Care Research Health Protection Research Unit (NIHR HPRU) in Blood Borne and Sexually Transmitted Infections at University College London in partnership with UKHSA, London, England, United Kingdom, **3** School of Health and Social Care, Edinburgh Napier University, Edinburgh, Scotland, United Kingdom, **4** Institute of Psychiatry, Psychology & Neuroscience at King's College London, England, United Kingdom, **5** Institute for Global Health, University College London, London, England, United Kingdom

* matthew.hibbert@ukhsa.gov.uk

**Data Availability Statement:** Sharing of public health data is subject to strict regulation due to the risk of disclosure and therefore cannot be made publicly accessible. Data can be made available

## Abstract

People living with severe mental illness have an increased prevalence of bloodborne viruses (BBVs) such as hepatitis B (HBV) and hepatitis C viruses (HCV), and HIV. To help improve the physical health of people living with severe mental illness, we aim to understand associations with BBV testing and treatment provision among those tested in mental healthcare settings in England. HBV surface antigen [HBsAg], antibody HCV [anti-HCV] and HIV testing and demographic information pertaining to people tested in mental health settings in England were extracted from a BBV testing dataset. Records pertaining to individuals diagnosed with HCV or HIV were linked to treatment datasets. Multivariable logistic regression analyses were used to understand demographic associations with test positivity for each BBV. Between 2015–2021, 18,221 people tested for a BBV in a mental health setting (56% male, 71% White ethnicity), 90% of whom were in inpatient care. Testing positive for HBsAg, anti-HCV and HIV was 1.1% (95%CI: 0.93–1.26%), 4.3% (4.00–4.63%) and 1.1% (0.92–1.25%) respectively. In multivariable analyses, women had reduced odds of testing positive for anti-HCV and HIV compared to men. Being of Asian, Black, or another ethnicity was associated with increased odds of testing HBsAg positive and Black ethnicity was associated with a positive HIV test result compared to White ethnicity. White ethnicity was associated with testing anti-HCV test positive compared to all other ethnicities. Half (344/708) of those who were anti-HCV positive would have benefitted from treatment (HCV-RNA positive), of which 58% received treatment. HIV treatment (96%) and viral suppression (94%) after testing in mental healthcare was high. To improve the physical health of people living with mental health conditions and to aid England's hepatitis elimination and HIV transmission goals, opt-out testing for BBVs may be beneficial to reduce health inequalities among people experiencing mental illness.

upon reasonable request to Hepatitis@ukhsa.gov.uk.

**Funding:** The research was funded by the National Institute for Health and Care Research Health Protection Research Unit (NIHR HPRU) in Blood Borne and Sexually Transmitted Infections at University College London in partnership with UK Health Security Agency (UKHSA) (MH, RS, and CS). The funders had no role in study design, data collection and analysis, decision to publish, or preparation of the manuscript.

**Competing interests:** The authors have no competing interests to declare

## Introduction

Severe mental illness includes psychiatric disorders (e.g. schizophrenia, schizoaffective disorder) and mood disorders (e.g. bipolar disorder), which are long-term and impair physical, occupational and social functioning [1,2]. People with mental health problems, including those with severe mental illness, have worse physical health compared to the general population [3]. However, many of the studies that have described physical health in this group have focussed on cardiovascular health with less attention having been paid to blood-borne viruses (BBVs) [4]. In London, HIV prevalence among people attending a mental healthcare service was 2.5 times greater than that in the local population [5]. A systematic review of HBV, HCV and HIV prevalence among people with severe mental illness found a higher prevalence of these BBVs compared to the general population in low prevalence regions, such as Europe, the UK, and the USA, which was not observed in higher prevalence regions such as Africa and southeast Asia [2]. Another international systematic review found a higher pooled estimate of HBV and HCV prevalence among men experiencing severe mental illness compared to women but interestingly found a lower pooled prevalence of HIV [6].

Severe mental illness has been associated with behaviours that are associated with BBV acquisition, such as inconsistent condom use, multiple sexual partners, and transactional sex [7], as well as injecting drug use [8,9]. In addition to severe mental illness, depression and anxiety have been associated with multiple sexual partners among women [10,11], and severe depressive symptoms have been associated with an increased likelihood of sharing injecting equipment compared to mild or moderate depressive symptoms [12].

In England, an offer of BBV testing is currently recommended for those experiencing severe mental illness who are perceived to be at risk, although what constitutes as 'at risk' is not specified [13]. If we are to continue to reduce morbidity and mortality associated with these BBVs for people experiencing severe mental illness, there is a need to understand the patterns of BBV testing and treatment provision for people receiving mental healthcare. Additionally, England has committed to targets of eliminating hepatitis as a public health threat [14,15], and achieving zero new HIV transmissions and HIV-related deaths, both by the year 2030 [16]. Therefore, BBV testing in mental healthcare may not only help reduce physical health inequalities experienced by people with mental health problems, but also aid the achievement of wider public health targets.

The aim of this study is to describe (i) the patterns of testing for HBV, HCV, and HIV in mental healthcare settings, (ii) the factors associated with test positivity following testing in these settings, and (iii) treatment provision among those who are diagnosed with these BBVs.

## Methods

### Ethical statement

Ethical approval was not required for this analysis as laboratory diagnosis data are collated and processed by UKHSA as part of surveillance of HBV, HCV and HIV infection and disease. These data collections are covered by Health Service (Control of Patient Information) Regulations 2002 (regulation 3) which makes provisions for the recognition, control and prevention of communicable diseases and other risks to public health. UKHSA's Caldicott Advisory Panel approved the linkage between SSBBV and HARS using pseudonymised information to ensure that the linked dataset could not be used to reveal anyone's identity (approval reference: CAP-2022-10). Data were accessed on the 13th of June 2023 for research purposes. Identifying information was used for linkage of participants' records between databases but not retained in the final dataset.

## Procedure

The sentinel surveillance system for blood borne viruses (SSBBV) collates BBV testing from twenty-three participating laboratories in England, which are estimated to cover approximately 40% of the English population [17]. Data collection methods for SSBBV have been described previously [18]. Testing data and demographic information for people aged 15 years or older tested for hepatitis B surface antigen (HBsAg), HCV antibody (anti-HCV), or HIV in mental healthcare settings between 2015–2021 were extracted from SSBBV. A mental healthcare setting test was determined by hospital, ward and community care specific information reported from laboratories processing the blood tests. Mental healthcare settings were independently coded as either inpatient or community by two researchers, where disagreements were resolved between the two researchers. Whether someone had previously received a negative tested in a prison or in a drug service was included in analyses due to a higher prevalence of BBVs, in particular HIV and HCV, in these populations [14,19]. Previous positive tests were excluded as these individuals would also test positive when retested in mental health settings.

**HBV.** Whether a person had an acute or chronic HBV at the time of their first positive test in a mental healthcare setting was established for people who tested HBsAg positive, which is a marker of current infection. Those with an antibody to HBV core antigen immunoglobin (anti-HBc IgM positive) were considered to have acute HBV. Those who received a negative anti-HBc IgM result after a positive HBsAg positive result were assumed to be chronic and those who did not have a linked anti-HBc IgM test had unknown acute or chronic status. Records from those who were HBsAg positive were also linked to the Hospital Episode Statistics (HES) dataset to obtain information on hepatocellular carcinoma (HCC) and end-stage liver disease (ESLD) outcomes (2012–2023 data included) [20].

**HCV.** For people who tested anti-HCV positive, follow-up HCV-RNA tests from SSBBV were extracted where available to establish those with current HCV. Records pertaining to people who tested HCV-RNA positive were linked to the HCV treatment database (2015–2022), which contains all HCV treatments in England for engagement in treatment services, treatment initiation, sustained virologic response (SVR; HCV clearance) and HCV reinfection information. The procedure of defining an HCV reinfection based on this linked dataset has been published previously [21], but briefly reinfection was defined as a positive HCV-RNA result from a sample taken ≥196 days after treatment initiation, where there was evidence of an SVR in the initial treatment period. Linkage was conducted based on NHS number, surname, soundex, date of birth, and gender.

Data pertaining to individuals testing positive for either HBsAg or anti-HCV were also linked to a national laboratory reporting dataset for individuals diagnosed with HBV or HCV to understand whether the test conducted in a mental health setting was a new diagnosis.

**HIV.** Linkage to the HIV and AIDS Reporting System (HARS) was undertaken for people with a positive HIV test result in mental healthcare settings to obtain additional demographic and treatment information (linkage to HIV clinical services, antiretroviral treatment (ART) initiation, likely mode of acquisition of HIV, and HIV-RNA (viral load). Linkage was conducted using clinical identifiers, soundex, date of birth, region, and gender. Due to potential delays across reporting systems, any test reported in the 7 days prior to the mental health test was grouped as the same episode of care and any positive test in that period was counted as a diagnosis in a mental health setting. Those previously diagnosed with HIV were identified through a positive HIV test more than 7 days prior to the test in the mental healthcare setting. Prior (within 12 months) ART use and viral load were obtained for those who had been diagnosed with HIV at least 12 months prior to their mental healthcare setting test. The nearest viral load in the 12 months after the mental healthcare setting test was also obtained, as well as

the person's last reported ART status and viral load post mental healthcare setting test. Viral suppression was classified as <200 copies/ml.

### Statistical analysis

The number of individuals tested for each virus and the test positivity are presented by year. If an individual tested positive, the individual was no longer included in the denominators for testing/positivity in subsequent years. If an individual had multiple tests in one year, and one of these was positive, the positive result was included for that year. Statistical analyses were conducted using RStudio (version 2023.03.0+386). Univariable and multivariable logistic regression analyses were used to assess the association of demographic factors as well as previous testing behaviours with test positivity for individuals tested for HBsAg, anti-HCV, and HIV separately. A parsimonious model was used, whereby factors which demonstrated an association (p<0.1) with the outcome in univariable analyses were included in the multivariable model, but only factors where p<0.5 in the multivariable were retained. A person's first test in the time period was used for analyses.

## Results

Over the period 2015–2021, a total of 18,211 individuals received at least one test for either HBsAg, anti-HCV or HIV in mental healthcare, with 24,277 tests undertaken in total (median 1 test per person, range: 1–14). Over 90% (22,214/24,277) of tests were conducted in inpatient settings. The majority of people were tested for all three BBVs when tested (72%, 17,373/24,277), with 21,412/24,277 (88%), 21,575/24,277 (89%) and 20,451/24,277 (84%) tested for HBsAg, anti-HCV and HIV respectively. The median age of individuals at their first BBV test in the time period was 38 years (inter-quartile range 28–51 years), 56% (10,095/18,211) were male, and 71% (12,887/18,211) were of White ethnicity. A summary of findings relating to HBV, HCV and HIV testing and treatment can be seen in Fig 1A–1C respectively.

### HBV

**HBsAg testing between 2015 and 2021.** The number of people tested for HBsAg in mental healthcare settings increased between 2015 (N = 2,370) and 2016 (N = 2,980), and then remained steady between 2016 and 2021 (average 2829 tests per year) (Fig 2A). Test positivity decreased over time from 1.4% (34/2,370) in 2015 to 0.7% (21/2916) in 2021.

**Associations with a positive HBsAg test result.** In total, 177 out of 16,304 people (1.1%, 95%CI: 0.93–1.26%) tested HBsAg positive. In univariable logistic regression analyses (Table 1), a positive test result was more likely in those of non-white ethnicity and in those in the 25-34-year age group, with no difference in test positivity between men and women. Test positivity rates did not differ significantly by region, although analyses were limited by lack of power. In multivariable analyses, both factors remained significantly associated with test positivity, with people of all non-white/unknown ethnicities being more likely to test positive than those of White ethnicity and those aged 15–24 years being less likely to test positive than those aged 25–34 years.

**HBV diagnosis and outcomes.** Just under half of those that tested positive for HBsAg (82/177, 46%) had a previous diagnosis and two tested anti-HBc IgM positive (1.2%, 2/171), indicating acute HBV infection. Five of those testing positive for HBsAg (3%) had a hospital admission associated with an HCC diagnosis and 5 (3%) had an admission associated with ESLD diagnosis (3 of these people had both an HCC and ESLD diagnosis).

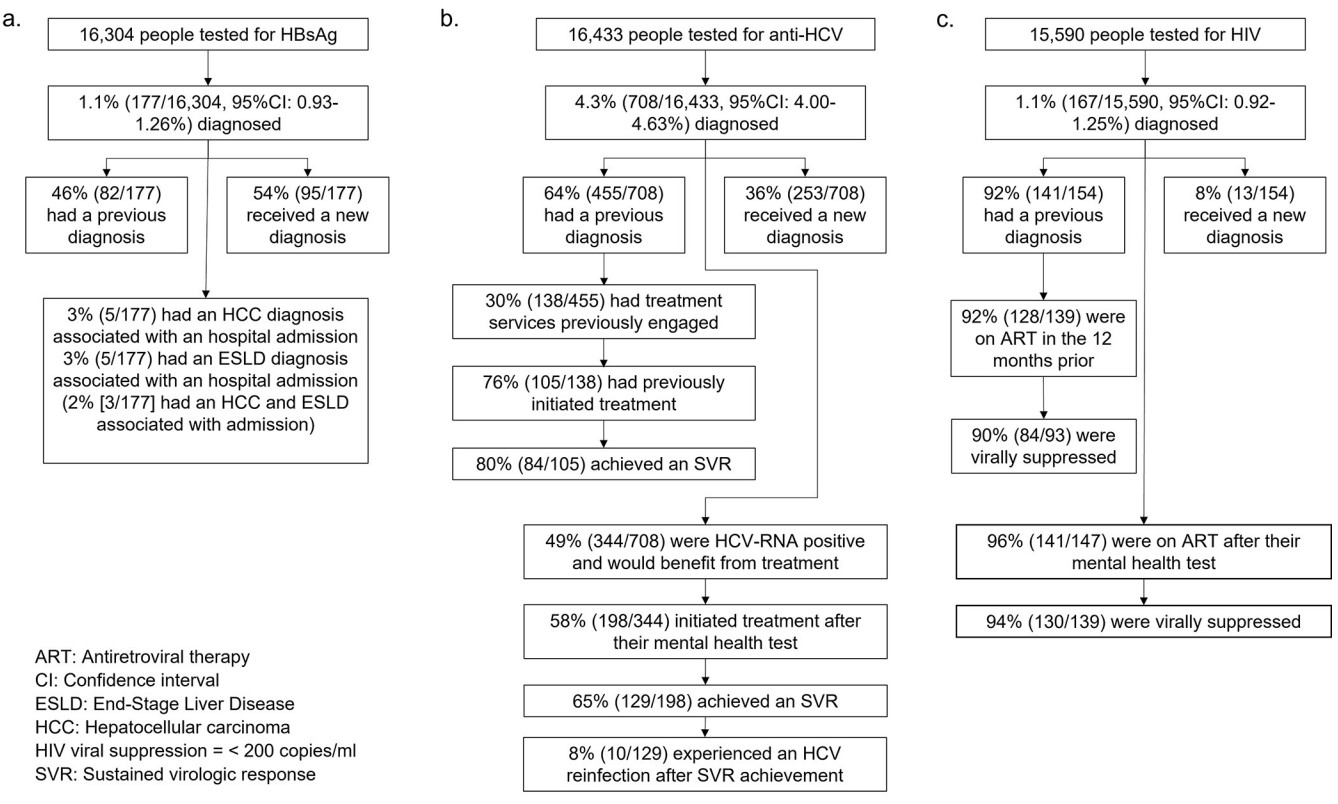

**Fig 1.** Overview of hepatitis B surface antigen (HBsAg) (a.), antibody hepatitis C (anti-HCV) (b.), and HIV (c.) testing and treatment in mental health settings in England, between 2015–2021.

## HCV

**Anti-HCV testing between 2015 and 2021.** The number of people tested for anti-HCV in mental healthcare settings increased between 2015 (N = 2,374) and 2016 (N = 3,107), and then steadily decreased between 2016 and 2020 (N = 2,571), with an increase in testing in 2021 (N = 2,811) (Fig 2B). Test positivity decreased from 5.7% (135/2,374) in 2015 to 2.6% (65/2,811) in 2021.

**Associations with a positive anti-HCV test result.** In total, 708/16,433 people tested positive for anti-HCV (4.3%, 95%CI: 4.00–4.63%). In univariable logistic regression analyses

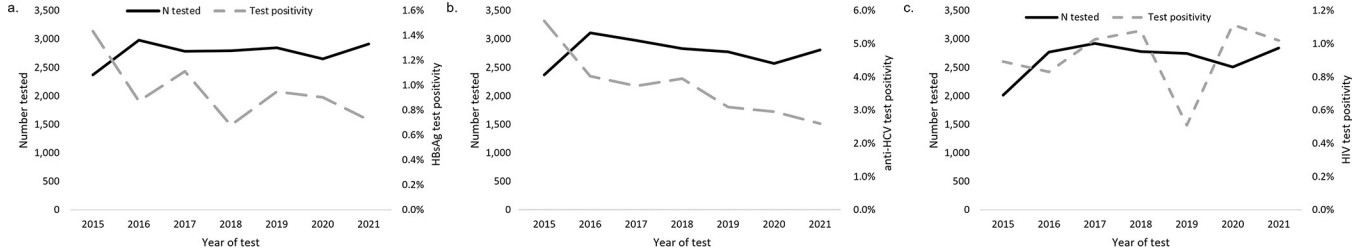

**Fig 2.** Number of individuals tested and test positivity for a. hepatitis B surface antigen (HBsAg), b. antibody HCV (anti-HCV), and c. HIV among people tested in mental health settings in England between 2015–2021. Note: Once an individual tested positive, the individual was no longer included for the number tested or test positivity for subsequent years. If an individual had multiple tests in one year, and one of these were positive, the positive result was included for that year.

**Table 1. Univariable and multivariable analyses for sociodemographic factors associated with hepatitis B surface antigen (HBsAg) test positivity among people tested in mental health settings between 2015–2021.**

| | HBsAg test result | | | OR(95%CI) | p-value | aOR(95% CI) | p-value |
|---|---|---|---|---|---|---|---|
| | Negative (N = 16,127) | Positive (N = 177) | Row % | | | | |
| **Gender** | | | | | 0.267 | | |
| Male | 9,081 | 107 | 1.2 | ref. | | - | |
| Female | 7,046 | 70 | 1.0 | 0.84 (0.62, 1.14) | | - | |
| **Ethnicity** | | | | | <0.0001 | | <0.0001 |
| White | 11,453 | 51 | 0.4 | ref. | | ref. | |
| Asian | 1,937 | 53 | 2.7 | 6.14 (4.17, 9.07) | | 6.31 (4.26, 9.34) | |
| Black | 1,334 | 45 | 3.3 | 7.58 (5.04, 11.35) | | 7.89 (5.22, 11.88) | |
| Mixed/Other | 563 | 20 | 3.4 | 7.98 (4.17, 9.07) | | 8.15 (4.71, 13.59) | |
| Unknown | 840 | 8 | 0.9 | 2.14 (0.94, 4.27) | | 2.20 (0.96, 4.40) | |
| **Age group** | | | | | 0.006 | | 0.003 |
| 15–24 | 2,608 | 15 | 0.6 | 0.45 (0.25, 0.78) | | 0.44 (0.24, 0.76) | |
| 25–34 | 4,410 | 56 | 1.3 | ref. | | ref. | |
| 35–49 | 4,895 | 68 | 1.4 | 1.09 (0.77, 1.57) | | 1.24 (0.87, 1.78) | |
| 50–64 | 3,007 | 29 | 1.0 | 0.76 (0.48, 1.18) | | 0.94 (0.59, 1.47) | |
| > = 65 | 1,207 | 9 | 0.7 | 0.59 (0.27, 1.13) | | 1.01 (0.46, 1.96) | |
| **Previous negative HBsAg test in a drug service** | | | | | | | 0.144 |
| No | 15,787 | 176 | 1.0 | ref. | | - | |
| Yes | 340 | 1 | 0.3 | 0.26 (0.02, 1.18) | 0.091 | - | |
| **Previous negative HBsAg test in a prison service** | | | | | | | |
| No | 15,794 | 177 | 1.1 | - | | - | |
| Yes | 333 | 0 | 0 | - | | - | |

(Table 2), test positivity rates varied significantly by gender, ethnicity, age group, whether the person had a previous negative test in a drug service, and region. In multivariable analyses, all factors remained significantly associated with test positivity. In particular: women were less likely to test positive than men; those of all non-White or unknown ethnicities were less likely to test positive than those of White ethnicity; test positivity generally increased with older age, other than in those aged ≥65 years where test positivity reduced; those who had previously tested negative in a drug service were more likely to test positive; and people tested in regions outside of London were more likely to have a positive test than those tested in London.

**HCV diagnosis and outcomes.** Among those with a positive anti-HCV test, nearly two-thirds (64%, 455/708) had a previous HCV diagnosis; 30% (138/455) of these people had been referred to HCV treatment prior to having their test in a mental healthcare setting and three-quarters (76%, 105/138) had initiated HCV treatment prior to their test in a mental healthcare setting, 80% of whom [84/105] had achieved an SVR (HCV clearance).

Nearly half of those who were anti-HCV positive were HCV-RNA positive (49%, 344/708; 2.1% of all those tested for anti-HCV [344/16,433]), which indicates a need for treatment; 16% (56/344) of whom had previously engaged with HCV treatment services. Three-fifths of those who were HCV-RNA positive (58%, 198/344) initiated treatment at some point after their test in the mental healthcare setting and 65% (129/198) achieved an SVR.

There were 13 individuals who experienced HCV reinfection on or after their mental healthcare setting test, three of whom were diagnosed with an HCV reinfection when tested in a mental healthcare setting.

**Table 2. Univariable and multivariable analyses for sociodemographic factors associated with hepatitis C antibody (anti-HCV) test positivity among people tested in mental health settings between 2015–2021.**

| | anti-HCV test result | | | OR(95%CI) | p-value | aOR(95% CI) | p-value |
|---|---|---|---|---|---|---|---|
| | Negative (N = 15,725) | Positive (N = 708) | Row % | | | | |
| **Gender** | | | | | <0.0001 | | <0.0001 |
| Male | 8926 | 503 | 5.3 | ref. | | ref. | |
| Female | 6799 | 205 | 2.9 | 0.54 (0.45, 0.63) | | 0.56 (0.47, 0.66) | |
| **Ethnicity** | | | | | <0.0001 | | <0.0001 |
| White | 11013 | 628 | 5.4 | ref. | | ref. | |
| Asian | 1962 | 33 | 1.7 | 0.29 (0.20, 0.41) | | 0.31 (0.21, 0.43) | |
| Black | 1369 | 5 | 0.4 | 0.06 (0.02, 0.14) | | 0.07 (0.02, 0.14) | |
| Mixed/Other | 561 | 11 | 1.9 | 0.34 (0.18, 0.60) | | 0.35 (0.18, 0.62) | |
| Unknown | 820 | 31 | 3.6 | 0.66 (0.45, 0.94) | | 0.74 (0.50, 1.05) | |
| **Age group** | | | | | <0.0001 | | <0.0001 |
| 15–24 | 2595 | 13 | 0.5 | 0.16 (0.09, 0.28) | | 0.17 (0.09, 0.29) | |
| 25–34 | 4296 | 132 | 3.0 | ref. | | ref. | |
| 35–49 | 4669 | 356 | 7.1 | 2.48 (2.03, 3.05) | | 2.34 (1.91, 2.89) | |
| 50–64 | 2899 | 191 | 6.2 | 2.14 (1.71, 2.69) | | 1.99 (1.59, 2.51) | |
| > = 65 | 1266 | 16 | 1.2 | 0.41 (0.23, 0.67) | | 0.37 (0.21, 0.60) | |
| **Previous negative anti-HCV test in a drug service** | | | | | <0.0001 | | <0.0001 |
| No | 15497 | 671 | 4.2 | ref. | | ref. | |
| Yes | 228 | 37 | 14.0 | 3.75 (2.59, 5.28) | | 2.81 (1.92. 4.01) | |
| **Previous negative anti-HCV test in a prison service** | | | | | 0.686 | | |
| No | 15449 | 697 | 4.3 | ref. | | - | |
| Yes | 276 | 11 | 3.8 | 0.88 (0.45, 1.54) | | - | |

## HIV

**HIV testing between 2015 and 2021.** The number of people tested for HIV in mental healthcare settings increased between 2015 (N = 2,014) and 2017 (N = 2,922) and then steadily decreased between 2017 and 2020 (N = 2,510), with an increase in testing in 2021 similar to testing levels seen in 2017 (N = 2,842) (Fig 2C). HIV test positivity ranged between 0.8%-1.1% between 2015–2021, apart from a dip seen in 2019 where test positivity was 0.5% (14/2,747).

**Associations with a positive HIV test result.** In total, 167/15,590 people had a positive HIV test result within a mental healthcare setting (1.1%, 95%CI: 0.92–1.25%). In univariable logistic regression analyses (Table 3), test positivity was associated with gender, ethnicity and age. The vast majority of positive tests were from people tested within mental healthcare settings in London. In multivariable analyses, these factors remained significantly associated with test positivity, with women being less likely to have a positive test than men, those of Black or unknown ethnicities being more likely to have a positive test than those of White ethnicity, and test positivity rates generally being highest in those aged 35–49 and 50–64 years.

**HIV diagnosis and outcomes.** Records pertaining to the majority of those who received a confirmatory HIV test in mental healthcare settings could be linked to a record in the HARS (92%, 154/167). Nearly three-quarters of those with linked records were men (73%, 122/154), two-fifths had been born in the UK (43%, 67/154) and just under 30% had been born in sub-Saharan Africa (29%, 44/154). Half of those with a linked HARS record were thought to have acquired HIV through sex between men (51%, 78/154), nearly two-fifths (38%, 59/154) had likely acquired HIV through sex between men and women, and the vast majority (92%, 141/154) had a previous diagnosis recorded prior to their mental healthcare setting test. Prior to

**Table 3. Univariable and multivariable analyses for sociodemographic factors associated with HIV test positivity among people tested in mental health settings between 2015–2021.**

| | HIV test result | | | OR(95%CI) | p-value | aOR(95% CI) | p-value |
|---|---|---|---|---|---|---|---|
| | Negative (N = 15,423) | Positive (N = 167) | Row % | | | | |
| **Gender** | | | | | <0.0001 | | <0.0001 |
| Male | 8576 | 121 | 1.4 | ref. | | ref. | |
| Female | 6847 | 46 | 0.7 | 0.48 (0.34, 0.66) | | 0.47 (0.33, 0.66) | |
| **Ethnicity** | | | | | 0.005 | | 0.003 |
| White | 10802 | 112 | 1.0 | ref. | | ref. | |
| Asian | 1824 | 13 | 0.7 | 0.69 (0.37, 1.18) | | 0.70 (0.38, 1.21) | |
| Black | 1361 | 25 | 1.8 | 1.77 (1.12, 2.69) | | 1.82 (1.15, 2.78) | |
| Mixed/Other | 560 | 2 | 0.4 | 0.34 (0.06, 1.09) | | 0.36 (0.06, 1.13) | |
| Unknown | 876 | 15 | 1.7 | 1.65 (0.92, 2.75) | | 1.85 (1.03, 3.10) | |
| **Age group** | | | | | <0.0001 | | <0.0001 |
| 15–24 | 2538 | 12 | 0.5 | 0.65 (0.32, 1.23) | | 0.65 (0.32, 1.23) | |
| 25–34 | 4241 | 31 | 0.7 | ref. | | ref. | |
| 35–49 | 4602 | 79 | 1.7 | 2.35 (1.56, 3.62) | | 2.37 (1.57, 3.65) | |
| 50–64 | 2856 | 41 | 1.4 | 1.96 (1.23, 3.16) | | 1.96 (1.23, 3.16) | |
| > = 65 | 1186 | 4 | 0.3 | 0.46 (0.14, 1.17) | | 0.50 (0.15, 1.27) | |
| **Previous negative HIV test in a drug service** | | | | | 0.181 | | |
| No | 15147 | 166 | 1.1 | ref. | | - | |
| Yes | 276 | 1 | 0.4 | 0.33 (0.02, 1.48) | | - | |
| **Previous negative HIV test in a prison service** | | | | | 0.371 | | |
| No | 15120 | 162 | 1.1 | ref. | | - | |
| Yes | 303 | 5 | 1.6 | 1.54 (0.54, 3.40) | | - | |

mental healthcare testing, 92% (128/139) were on ART and 90% (84/93) were virally suppressed; within the 12 months post mental healthcare testing 82% were virally suppressed, and at their last HIV care attendance 96% (141/147) were on ART and 94% (130/139) were virally suppressed.

## Discussion

This study aimed to understand HBV, HCV, and HIV testing in mental healthcare settings and subsequent engagement with treatment services for those diagnosed with a BBV. The proportion of people who had a positive test result was high across all BBVs. A higher proportion of new diagnoses were found among those tested for HBV (54%) compared to HCV (36%) and HIV (8%). Treatment initiation (58%) and treatment completion (65%) was low for those with reactive HCV-RNA (individuals who are eligible for treatment), highlighting a need to identify and remove barriers to HCV treatment provision for people experiencing mental health problems.

We found a higher prevalence of HBV, HCV, and HIV (1.1% for HBsAg and HIV, 2% for HCV-RNA) among people tested in mental healthcare settings compared to the general public (0.4%, 0.1% and 0.2% respectively) [14,16,22,23] and the declining trends in test positivity for HBsAg and anti-HCV are similar to what has been seen nationally [14,22]. This is also consistent with an international systematic review investigating BBVs among people living with severe mental illness [2]. Compared to emergency department testing in England, where HBsAg, HCV-RNA, and HIV test positivity was 1.1%, 0.2%, and 0.9% for HIV respectively [24], we found similar overall test positivity for HBsAg and HIV, but HCV-RNA test positivity

among mental health service attendees was ten times that which was found in emergency departments (2.1%). This represents a substantially larger proportion of people that would benefit from HCV treatment found though testing in mental health settings compared in emergency department BBV case-finding initiative. New diagnosis rates for HBV were similar to those seen following emergency department testing, but in this study, new diagnoses for HCV and HIV were lower (34% vs. 62% and 10% vs. 35% respectively). It is important to note that the vast majority (>90%) of individuals included were tested in inpatient settings and this makes up a small proportion (0.2%) of people in mental healthcare in England [25]. Whilst those in community/outpatient care may be a slightly different population, further research should establish whether BBV testing in all mental health care settings could provide a benefit to people living with mental illness. This is because behaviours that are associated with potential exposure to HBV, HCV, and HIV have been observed at an increased prevalence among people with severe mental illness and mental illness generally compared to the general population [7–12].

Similar to an international systematic review, men were more likely to have an HCV diagnosis than women, but contradictory to this systematic review, we also found higher rate of HIV diagnoses among men with no gender difference seen for HBV diagnoses [6]. Among the people living with HIV in this study, the proportion who acquired HIV through sex between men was similar to that among people seen for HIV care in England (51% vs. 45%) [23]. It therefore appears that MSM living with HIV were not disproportionately seen in mental healthcare services. Being previously tested in a drug service, which may indicate previous drug use, was associated with anti-HCV test positivity in this study and people who inject drugs represent a large proportion of people seen for HCV care in England [14].

The likelihood of a positive test result also varied by ethnicity and virus: those of any non-White ethnicity were more likely to have a positive HBsAg test result, those of White ethnicity were most likely to test positive for anti-HCV, and those of Black ethnicity were more likely to test positive for HIV. In low HBV prevalence countries like the UK, HBV is most common in people who have migrated from high prevalence countries [26,27], so the test positivity variations by ethnicity found in this study are expected. However, further work is needed to understand possible routes of acquisition, as if HBV is acquired more recently, then preventative measures like HBV vaccination may be beneficial for people with mental illness. The association between White ethnicity and a positive HCV result is similar to what is seen nationally, due to the majority of people diagnosed with HCV and seen for HCV care are White British people who inject or have previously injected drugs [21,28]. Additionally, people of Black ethnicity represent around one-third of people seen for HIV care in England [23]. Therefore, it appears as though whilst there may be a higher prevalence of BBVs among people with mental illness, the variations in test positivity across different ethnicities is broadly representative of those diagnosed nationally.

HIV treatment coverage and viral suppression was high and in line with UN 90-90-90 targets (90% of people diagnosed on treatment and 90% on treatment virally suppressed) [29], although the proportion of people living with HIV on treatment and virally suppressed (92% and 90%) prior to mental healthcare setting testing were both slightly below the national rates of 98% for each [23]. Treatment and viral suppression rates were, however, higher post-mental healthcare setting testing, which may indicate a positive benefit of the receipt of mental healthcare treatment on individuals' health in relation to HIV. People living with HIV experience poorer mental health than the general population [30,31] and ensuring HIV treatment is continued in times of poorer mental health will help reduce possible worse physical health in this population.

A limitation of this analysis is that the uptake of BBV treatment after receiving a positive test in a mental healthcare setting could only be described for HCV and HIV as no national

dataset for HBV treatment exists. Thus, to understand the HBV-related health of this group, linkage to hospital attendances was undertaken to record the use of hepatitis-related care. From the emergency department BBV testing in England, treatment services were poorly engaged with those who were diagnosed [24], and in this study treatment service provision to people who would benefit from HCV treatment could have been improved. Therefore, it is important that any new testing initiatives for people with mental health conditions have well established care pathways if health and wellbeing related to BBVs in this population is to be improved. Another limitation of this analysis is that we could not establish how representative this testing was of the wider population of people accessing mental healthcare services, nor determine the proportion of people attending these services who were tested for BBVs. Whilst we were able to determine whether the testing came from inpatient or community services, we could not reliably state whether these services were specific to people with severe mental illnesses or also provided other types of mental health care (e.g. dementia). Regardless, BBV prevalence appeared to be higher in this study than the general population and testing for BBVs in different types of mental health setting may be needed to reduce this health inequality.

In conclusion, this analysis highlights an unmet healthcare need regarding BBVs among people receiving mental healthcare in England, with particular reference to HCV. Care pathways from mental health services to hepatitis and HIV care may need to strengthened and barriers to care identified, understood, and removed. To aid the support of testing for HBV, HCV and HIV, specific guidance around BBV testing in mental healthcare may be needed and opt-out testing in severe mental health care should be considered to improve the health and wellbeing of people living with mental health conditions and BBVs.

## Acknowledgments

We acknowledge members of the NIHR HPRU in BBSTI Steering Committee: Professor Caroline Sabin (HPRU Director), Dr John Saunders (UKHSA Lead), Professor Catherine Mercer, Dr Hamish Mohammed, Professor Greta Rait, Dr Ruth Simmons, Professor William Rosenberg, Dr Tamyo Mbisa, Professor Rosalind Raine, Dr Sema Mandal, Dr Rosamund Yu, Dr Samreen Ijaz, Dr Fabiana Lorencatto, Dr Rachel Hunter, Dr Kirsty Foster and Dr Mamoona Tahir.

## Author Contributions

**Conceptualization:** Matthew Hibbert, Liz Hughes, Monica Desai, Caroline Sabin.

**Data curation:** Matthew Hibbert, Ruth Simmons, Peter Dearman, James Lester, Annabel Powell, Cuong Chau, Clare Humphreys.

**Formal analysis:** Matthew Hibbert, Ruth Simmons, Caroline Sabin.

**Funding acquisition:** Ruth Simmons, Monica Desai, Caroline Sabin.

**Investigation:** Matthew Hibbert.

**Methodology:** Matthew Hibbert, Ruth Simmons, Peter Dearman, James Lester, Annabel Powell, Cuong Chau, Clare Humphreys, Monica Desai.

**Project administration:** Matthew Hibbert.

**Resources:** Matthew Hibbert, Margaret Heslin.

**Supervision:** Ruth Simmons, Liz Hughes, Margaret Heslin, Monica Desai, Caroline Sabin.

**Validation:** Peter Dearman.

**Writing – original draft:** Matthew Hibbert, Ruth Simmons, Peter Dearman, James Lester, Annabel Powell, Monica Desai, Caroline Sabin.

**Writing – review & editing:** Matthew Hibbert, Ruth Simmons, Peter Dearman, James Lester, Annabel Powell, Cuong Chau, Clare Humphreys, Liz Hughes, Margaret Heslin, Monica Desai, Caroline Sabin.

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
