## [Decision Letter · Decision Letter 0]

7 May 2024

PMEN-D-24-00050

Testing for Hepatitis B and C virus and HIV in mental healthcare settings and engagement in care in England between 2015-2021

PLOS Mental Health

Dear Dr. Hibbert
Matthew,

Thank you for submitting your manuscript to PLOS Mental Health. After careful consideration, we feel that it has merit but does not fully meet PLOS Mental Health’s publication criteria as it currently stands. Therefore, we invite you to submit a revised version of the manuscript that addresses the points raised during the review process.

EDITOR: Be sure to:

Revise your title to "Testing for Hepatitis B and C virus and HIV in mental healthcare settings in England between 2015-2021". The other words in the title look redundant and make your title not so easy to understand.Add appropriate and complete sub-sections headings for your results. The present sub-section headings are fairly incomplete.Revise your discussion as per advise of the reviewer.

Please submit your revised manuscript by **6th June 2024.** If you will need more time than this to complete your revisions, please reply to this message or contact the journal office at mentalhealth@plos.org. Please include the following items when submitting your revised manuscript:

We look forward to receiving your revised manuscript.

Kind regards,

Kizito Omona, PhD

Academic Editor

PLOS Mental Health

Journal Requirements:

1. Please amend your detailed online Financial Disclosure statement. This is published with the article. It must therefore be completed in full sentences and contain the exact wording you wish to be published.

a) State the initials, alongside each funding source, of each author to receive each grant. For example: "This work was supported by the National Institutes of Health (####### to AM; ###### to CJ) and the National Science Foundation (###### to AM)."

2. Please update your online Competing Interests statement. If you have no competing interests to declare, please state: “The authors have declared that no competing interests exist.”

3. In the online submission form, you indicated that [Insert text from online submission form here]. 

a) In a public repository, 

b) Within the manuscript itself, or 

c) Uploaded as supplementary information.

Additional Editor Comments (if provided):

Title: Delete "and engagement in care" from the title. It looks a redundant statement, making the title not so easy to understand. And so, it is better to maintain your title as "Testing for Hepatitis B and C virus and HIV in mental healthcare settings in England between 2015-2021"

Results: Kindly expound on the respective sub-section headings. Those headings should match with the respective contents under the headings.

Reviewers' comments:

Reviewer's Responses to Questions

**Comments to the Author**

1. Does this manuscript meet PLOS Mental Health’s publication criteria? Is the manuscript technically sound, and do the data support the conclusions? The manuscript must describe methodologically and ethically rigorous research with conclusions that are appropriately drawn based on the data presented.

Reviewer #1: Yes

Reviewer #2: Yes

2. Has the statistical analysis been performed appropriately and rigorously?

Reviewer #1: Yes

Reviewer #2: Yes

3. Have the authors made all data underlying the findings in their manuscript fully available (please refer to the Data Availability Statement at the start of the manuscript PDF file)?

Reviewer #1: Yes

Reviewer #2: Yes

4. Is the manuscript presented in an intelligible fashion and written in standard English?

Reviewer #1: Yes

Reviewer #2: Yes

5. Review Comments to the Author

Reviewer #1: In the discussion I missed presenting and discussing the possible reasons that may have interfered with the increased prevalence rates (HBV, HCV and HIV) in people with serious mental illnesses. Just like this difference in race profile.

Reviewer #2: This study determines the Testing for Hepatitis B and C virus and HIV in mental healthcare settings and 2 engagement in care in England between 2015-2021. The study is well written and publishing it will open the eyes of other readers to concentrate on patients in mental health facilities and those suffering from mental health.

The authors should read through to correct minor mistakes.

6. PLOS authors have the option to publish the peer review history of their article (what does this mean?). If published, this will include your full peer review and any attached files.

**Do you want your identity to be public for this peer review?** For information about this choice, including consent withdrawal, please see our Privacy Policy.

Reviewer #1: **Yes: **JAKELINE RIBEIRO BARBOSA

Reviewer #2: No

---

## [Editor Report · Decision Letter 1]

20 Jun 2024

Testing for Hepatitis B and C virus and HIV in mental healthcare settings in England between 2015-2021

PMEN-D-24-00050R1

Dear Dr. Hibbert,

We are pleased to inform you that your manuscript 'Testing for Hepatitis B and C virus and HIV in mental healthcare settings in England between 2015-2021' has been provisionally accepted for publication in PLOS Mental Health.

Best regards,

Kizito Omona, PhD

Academic Editor

PLOS Mental Health

Thank you for addressing all the concerns raised